# One Dimensional Twisted Van der Waals Structures Constructed by Self-Assembling Graphene Nanoribbons on Carbon Nanotubes

**DOI:** 10.3390/ma15228220

**Published:** 2022-11-18

**Authors:** Kun Zhou, Liya Wang, Ruijie Wang, Chengyuan Wang, Chun Tang

**Affiliations:** 1Faculty of Civil Engineering and Mechanics, Jiangsu University, Zhenjiang 212013, China; 2Zienkiewicz Centre for Computational Engineering, Faculty of Science and Engineering, Bay Campus, Swansea University, Swansea SA1 8EN, UK; 3Key Laboratory for Intelligent Nano Materials and Devices of Ministry of Education, Nanjing University of Aeronautics and Astronautics, Nanjing 210016, China

**Keywords:** chiral carbon nanotube, graphene nanoribbon, self-assembly, twist angle

## Abstract

Twisted van der Waals heterostructures were recently found to possess unique physical properties, such as superconductivity in magic angle bilayer graphene. Owing to the nonhomogeneous stacking, the energy of twisted van der Waals heterostructures are often higher than their AA or AB stacking counterpart, therefore, fabricating such structures remains a great challenge in experiments. On the other hand, one dimensional (1D) coaxial van der Waals structures has less freedom to undergo phase transition, thus offer opportunity for fabricating the 1D cousin of twisted bilayer graphene. In this work, we show by molecular dynamic simulations that graphene nanoribbons can self-assemble onto the surface of carbon nanotubes driven by van der Waals interactions. By modifying the size of the carbon nanotubes or graphene nanoribbons, the resultant configurations can be controlled. Of particular interest is the formation of twisted double walled carbon nanotubes whose chiral angle difference can be tuned, including the 1.1° magic angle. Upon the longitudinal unzipping of such structures, twisted bilayer graphene nanoribbons can be obtained. As the longitudinal unzipping of carbon nanotubes is a mature technique, we expect the strategy proposed in this study to stimulate experimental efforts and promote the fast growing research in twistronics.

## 1. Introduction

Carbon nanotubes (CNTs) and graphene are among the most exciting low dimensional materials that possess extraordinary properties, such as ultrahigh strength, unique configuration dependent electronical conductivity and magnetic properties, etc. Such unusual properties make CNTs and graphene promising for applications in a wide range of areas [1,2,3,4,5,6,7,8,9]. For example, CNTs have been used for next generation high performance field effect transistors, ultrastrong artificial muscles, gigahertz (GHz) oscillators and so on, while graphene has been designed for solid lubricants, sea water desalination, and drug delivery, etc. [10,11,12,13,14].

Recent progress shows that by stacking one layer of graphene on top of another, the formed bilayer graphene exhibits more distinct properties than its monolayer counterpart, e.g., the open up of an electronic band gap [15]. Similarly, stacking two layers of different two-dimensional (2D) materials can form the so-called van der Waals (vdW) heterostructures, which host superior electronic properties that are promising for advanced devices [16,17,18,19,20,21]. Most strikingly, Cao et al. [18,19] showed that via twisting one layer of graphene by a certain angle (1.1°), called the magic angle, exotic electron correlation behavior occurs and the magic angle bilayer graphene exhibits superconducting property. This surprising finding has significantly boosted research interests of twistronics [20,21].

On the other hand, although twisted bilayer vdW heterostructures have shown great promise in electronic devices, their mechanical and thermal stability remains a pressing issue [22,23,24]. This is because commensurate stacking in 2D materials has the lowest total energy, while twisting naturally leads to an incommensurate state. For example, when subjected to sliding lateral forces, twisted graphene exhibits lower frictional responses, while untwisted graphene layers have the highest friction, meaning a weak external stimulus may easily cause the twisted vdW structures to restore to their original low energy state [25].

Interestingly, as graphene’s 1D counterpart, CNTs are generally regarded as rolled up of graphene layers and multi-walled CNTs (MWCNTs), and can be composed of co-axial single-walled CNTs (SWCNT) of different chiralities. The different combination of chiralities among inner and outer CNTs can also result in rich electronic states. Furthermore, several groups have shown that MWCNTs can be unzipped or partially unzipped to 1D graphene nanoribbons (GNRs) or GNR-CNT junctions [26,27,28,29,30,31,32,33,34,35,36]. Once unzipped, particularly when partially unzipped, the relative orientations of the different layers of the resultant nanostructures could be well controlled and consistent with the original 1D counterpart.

Motivated by this scenario, in this work we propose a strategy to fabricate 1D vdW heterostructure via self-assembly of GNR onto the surface of SWCNT. The size of the GNR and SWCNT can be used to tune the resultant heterostructure with the desired orientation, i.e., with a particular twist angle between the inner and outer layer. Upon partially unzipping the 1D vdW heterostructure, the resultant GNR bilayer is shown to exhibit a twisted moire pattern. This strategy is expected to provide new alternatives to synthesizing magic angle bilayer graphene, or other twisted phases of 2D materials.

## 2. Materials and Methods

The molecular dynamics (MD) simulations are performed using the Large-scale Atomic/Molecular Massively Parallel Simulator (LAMMPS) [37]. The general procedure in this package is to solve Newton’s equation of motion for atoms at particular conditions, using numerical methods. This technique has been widely used to study mechanical behavior of nanomaterials or structures. The Airebo potential [38] is used to describe interactions for hydrocarbon and it is well accepted that this potential is accurate for studying low dimensional carbon systems, and the Lennard-Jones (LJ) potential [39] is used to study the vdW interactions. We note that we have also tested the Kolmogorov-Crespi (KC) potential [40] and the interlayer potential (ILP) [41] to validate our results obtained via LJ potential and found the conclusions in this work remain unchanged. A constant temperature ensemble (NVT) is used in the simulations with Nose-Hoover thermostat, and the time step for the simulation is 1 femtosecond (fs). SWCNTs of various chirality and GNRs of various sizes are chosen to study the self-assemble process. The structures are visualized via the Open Visualization Tool (OVITO) for analysis [42].

## 3. Results

To validate the feasibility of the proposed fabrication process, we have run simulations by placing GNR of various sizes to the vicinity of SWCNTs to examine the dynamical behavior. Typical results are shown in Figure 1, a (44,6) SWCNT is chosen as the template, a GNR of size 13.6 × 10.0 nm^2^ is placed next to the SWCNT. After a few picoseconds (ps), the potential energy of the system displays a notable decreasing trend (see Figure 1d), due to the contact of the GNR with the SWCNT that generates strong vdW interaction. Next, the GNR bends around the SWCNT, thereby the potential energy shows slight increasement. Afterwards, owing to the increasing vdW interaction, the GNR completely wraps over the SWCNT outer wall, forming one dimensional vdW heterostructures, and the potential energy soon decreases to its minimum value at around 30 ps.

The wrapping behavior is observed in all the models we simulated, and the mechanism can be primarily attributed to the vdW interaction that overcomes the bending energy induced by wrapping. Interestingly, we find that the resultant configuration varies, however, depending on the size of the models we constructed, this opens up spaces to fine tune the resultant configurations. Taking the (44,6) SWCNT template as an example, three phases are observed, when the length of the GNR is as large as 70 nm, a longitudinal (L) wrapping is observed, forming multi-layered nanoscrolls. When the length is small, in-between 5 nm and 10 nm for example, helical (H) wrapping is achieved. Meanwhile, for an intermediate sized GNR, the edges of the helical structure attract to each other, and then make contact to form a tubular (T) structure. A widthwise (W) wrapping is also observed for the (49,0) SWCNT template, which has a similar size as the (44,6) SWCNT, the reason can be attributed to the fact that commensurate stacking between SWCNT and GNR can reduce the total potential energy of the system. Therefore, for chiral SWCNTs, H phases are preferred for small GNR, while for zigzag or armchair SWCNT templates, W phases are preferred. The observed phases are summarized in Figure 1e,f. It is worth noting that with experimental tools being quickly improved in recent decades, the assembly process described here can be achieved using various approaches. For example, experimental work by Tang et al. [27,31] has shown that by dispersing CNTs and GNRs separately in solutions, and then ejecting them simultaneously using coaxial syringes, they connect to the two solution sources. The assemble process can be achieved as the solution evaporates quickly. Although the length of the CNTs in the experiment is much larger than that used in simulations, the observed structures have been confirmed to agree well with MD predictions, indicating that the MD simulations have captured the main underlying mechanism. The functionality of such structures has also been tested in real devices.

One interesting feature observed in the T phase is that the size of the GNR does not have to be exactly 2πR, here R = r + 0.34 and r is the radius of the SWCNT, as depicted in Figure 2. This is because when L is somewhat smaller than 2πR, an H phase or W phase firstly form, afterwards the vdW interaction between the edges drives the GNR twisting around to approach each other as a path to reduce the potential energy. Once a bare edge GNR is utilized, the dangling atoms pair to their counterparts of the opposite edge, a seamless CNT can be formed and a double walled CNT (DWCNT) is therefore fabricated. Recent experimental works by Xiang et al. [43] reported that one dimensional vdW heterostructures can be formed via the CVD growth of BN or MoS_2_ nanotubes on SWCNT template; this process shares some similarities with ours, but the temperature is much higher in the experiment. More progress has also been recently made towards producing various 1D structures, such as the scrolls or ribbons in CNTs [44,45,46]. Simulation studies have also been reported by several groups [47,48,49], but the purpose and the design principles are different from our work here. Furthermore, as has been discussed above, if the size of the GNR is smaller than 2πR, a twisted outer wall of the SWCNT can be formed. This offers an exciting opportunity to fabricate 1D vdW heterostructures with the desired twisting, providing both the orientation of the inner and outer wall can be controlled. It is worth noting that not all the configurations from our simulations yield the perfect seamless DWCNTs, but the overall structural feature and twisting angle remain similar.

For the chirality of the inner wall, the synthesis of SWCNT with the desired chirality has shown enormous progress, while for the orientation of the outer wall, according to the model presented in Figure 2, a simple geometrical relationship can be found as:(1)L=2πR·cosθ
here *θ* is the twist angle of the GNR after wrapping. We have run simulations with GNRs of different length to wrap a template SWCNT, the resultant twist angle is plotted as a function of *L* in Figure 2, and the results agree well with the above relationship.

This relationship means that the chirality of the outer wall can be controlled by the size of the GNR. For example, to obtain the magic angle between the inner and outer wall of the DWCNT, one can take a zigzag SWCNT as the template, then, according to the size of the SWCNT and the relationship in Equation (1), a particular sized GNR can be chosen to wrap around the SWCNT to form a 1.1° twist angle. We demonstrate this scenario in our simulations, as a representative example shown in Figure 3, here the zigzag SWCNTs of different radius are chosen as a template and the GNRs of particular size that ensures *θ* in Equation (1) equal to 1.1° are used to wrap the SWCNTs. The resultant twisting angles are found to agree well with predictions. Shown in Figure 4 are alternative choices of SWCNT models that have a chiral angle of 1.1°; in this situation, the GNR size of exact values are needed to form perfect zigzag outer CNTs. Our simulation results in all yielded structures being well within expectation. It is worth noting, however, that when performing real experiments, it is not yet practical to differentiate zigzag and armchair edges of the GNR, therefore, either the armchair edges or the zigzag edges could move around to make contacts; thus, portions of the ultimate products could be a non-magical angle one dimensional vdW heterostructures.

We next proceed to study the stability of the as-formed structures under various temperature conditions, as representative results shown in Figure 5. The (44,6) SWCNT wrapped by the GNR of L = 13.4 nm is chosen for examination. In Figure 5a, the system is simulated at a temperature of 5 K, within 18 ps of simulation, the GNR wrapped around the SWCNT and then the bare edge atoms bond to each other, leading to the two-stage drop of the potential energy. At 40 ps, we steadily raise the temperature of the system to 300 K, and the potential energy follows a linear increment trend, indicating the structure does not experience dramatic phase transition or ductile failure. From 90 ps, the system is kept at 300 K, and no apparent change in potential energy and structural feature is observed, indicating the robustness of the as-formed 1D vdW heterostructure. A different temperature scheme is applied in Figure 5b, where the system is initially subjected to a temperature environment of 300 K and then steadily decreased to 5 K; interestingly, the assembled structure remains the same and temperature modulation does not influence the orientation of both walls of the DWCNT. In other words, the 1.1° magic angle is well preserved whether the assemble process is performed at 5 K or room temperature, as long as the size of the GNR and the SWCNT is well tailored. Considering that the fabrication of the magic angle bilayer graphene often occurs at higher temperature than the critical superconducting temperature, our results may provide promising routes to synthesizing the heterostructures of intriguing properties.

As mentioned above, several groups have experimentally demonstrated the fabrications of layered GNRs via the unzipping of MWCNTs; a recent work by Chen et al. [50] showed that the radial compression of large size MWCNTs can also break the chemical bonds of carbon at the highly stressed edges, leading to high quality GNRs for exotic electronic devices. We have conducted further simulations to mimic such experimental procedure and examine the structures of the unzipped twisted DWCNTs, as typical snapshots shown in Figure 6. After etching the top side of the (91,18)/(97,22) DWCNT, the structure starts to unfold and collapse on to the substrate, owing to the vdW interaction, full contact is finally achieved after 190 ps of simulation. The resultant bilayer graphene is shown in Figure 6e, where a clear moire pattern is observed and the twisting angle is 1.16°, slightly different from the chiral angle difference of the DWCNT (1.144°). We have also simulated partially unzipped DWCNTs and the conclusions remain unchanged.

## 4. Conclusions

In conclusion, we show by molecular dynamic simulations that graphene nanoribbons can be self-assembled to the surface of SWCNTs to form various heterostructures. If the size of the SWCNT and the GNR is well designed, the DWCNTs of particular twisting can be obtained; this offers a plausible platform for unzipping towards twisted bilayer graphene, including magic angle bilayer graphene. Upon unzipping, the twist angle for the resultant bilayer graphene remains almost unchanged. The obtained 1D vdW heterostructures is demonstrated to exhibit good structural stability under temperatures from 5 K to 300 K. This strategy can also be used to fabricate 1D vdW heterostructures composed of other elements, such as BN and MoS_2_. Future work towards fabricating multilayered vdW structures with precise twist should also be of high interest to this field, as recent experimental work has shown that such structures are also host interesting [51]. We expect our work to stimulate further experimental efforts in fabricating precisely controlled nanostructures towards high performance electronic devices.

## Figures and Tables

**Figure 1 materials-15-08220-f001:**
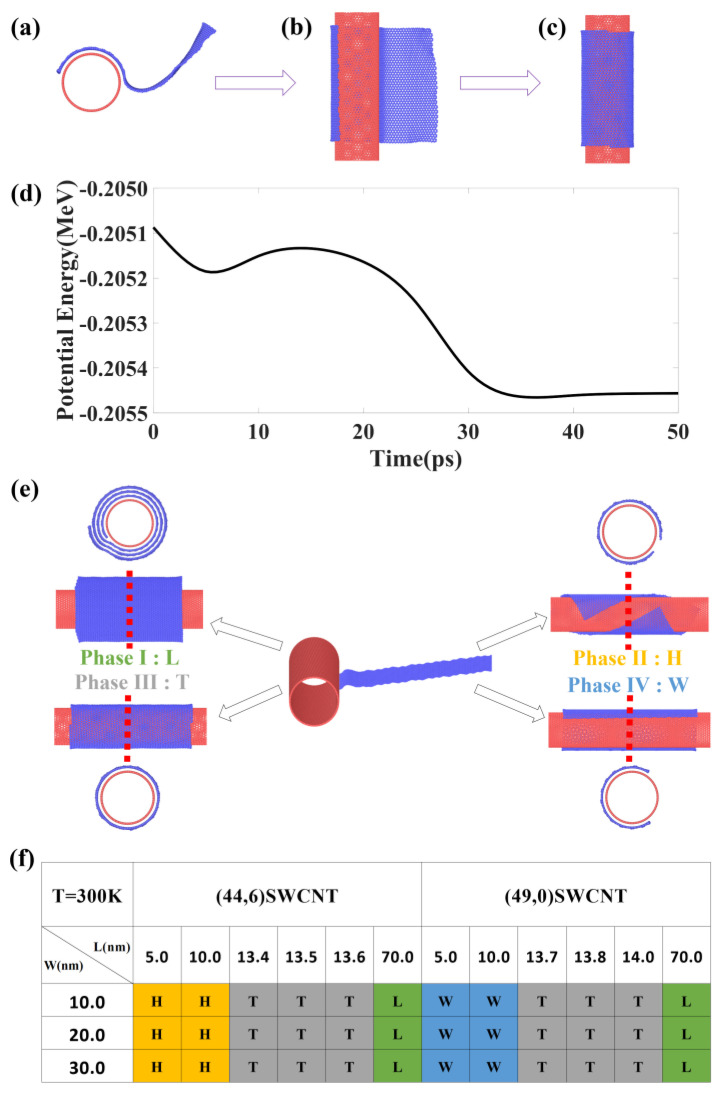
(**a**–**c**) Self-assembly of a graphene nanoribbon on the (44,6) SWCNT, the size of the GNR is 13.6 nm × 10.0 nm. (**d**) Corresponding potential energy variation. (**e**) Representative phases obtained from MD simulations. (**f**) Summarized phase diagram for the assembly associated with the size of the GNR.

**Figure 2 materials-15-08220-f002:**
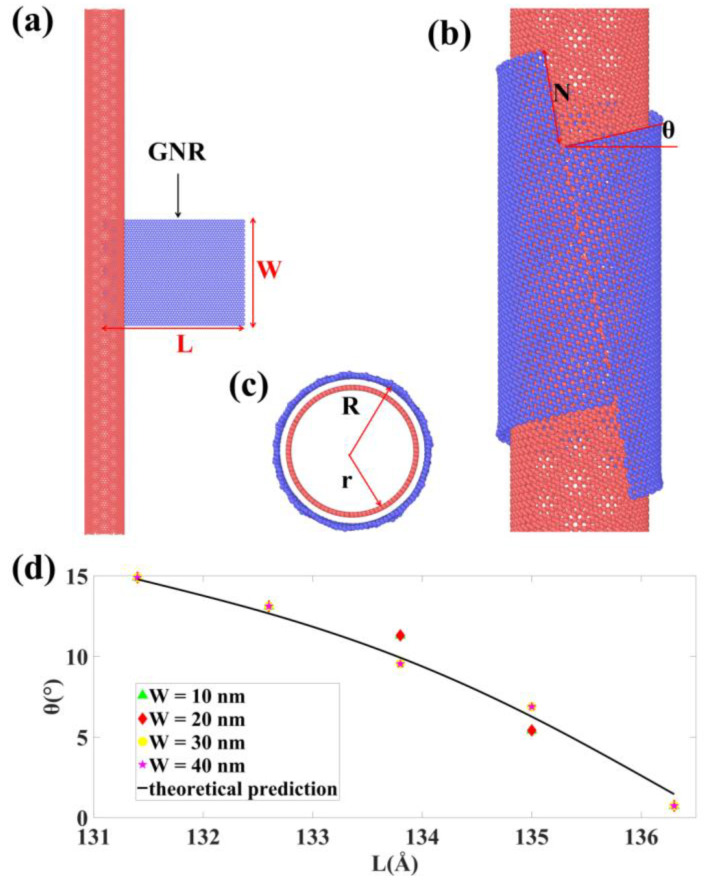
Geometrical relationships between helical angle and the sizes of SWCNTs and GNRs. The resultant *θ* from simulations agree well with predictions from Equation (1). (**a**) the starting model for simulations, (**b**,**c**) the final configuration after wrapping from side and top views and (**d**) the plot of θ as a function of L from both MD simulations and Equation (1).

**Figure 3 materials-15-08220-f003:**
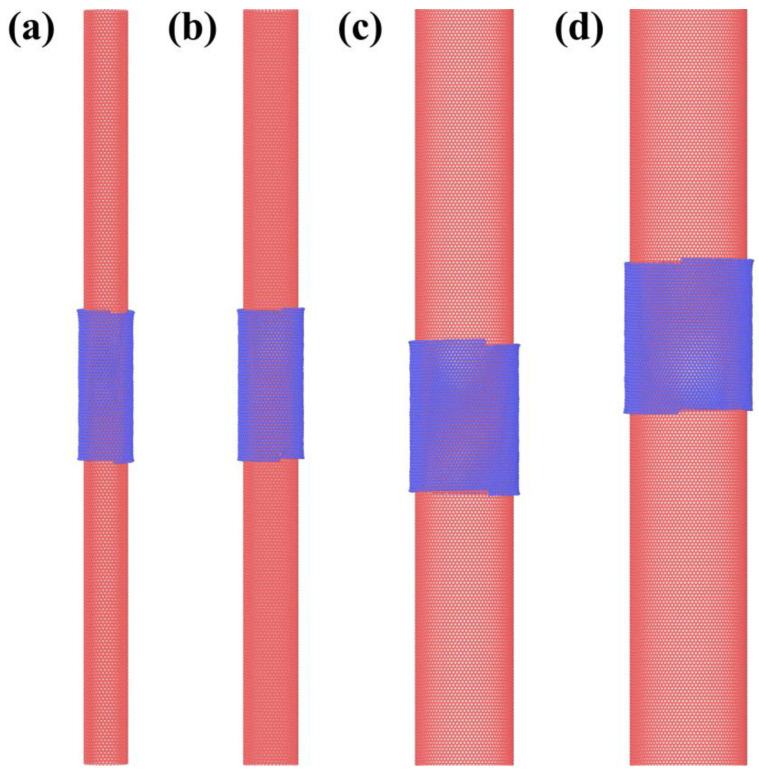
Self-assemble of GNR on the surface of zigzag SWCNTs. The chiral indexes of these SWCNTs from (**a**–**d**) are (36,0), (45,0), (82,0) and (98,0). The length of the GNRs are 10.99 nm, 13.21 nm, 10.99 nm and 22.31 nm, respectively. The obtained θ values are 1.1° for (**a**,**c**) and 0.93° for (**b**,**d**), both values correspond to superconducting magic angles.

**Figure 4 materials-15-08220-f004:**
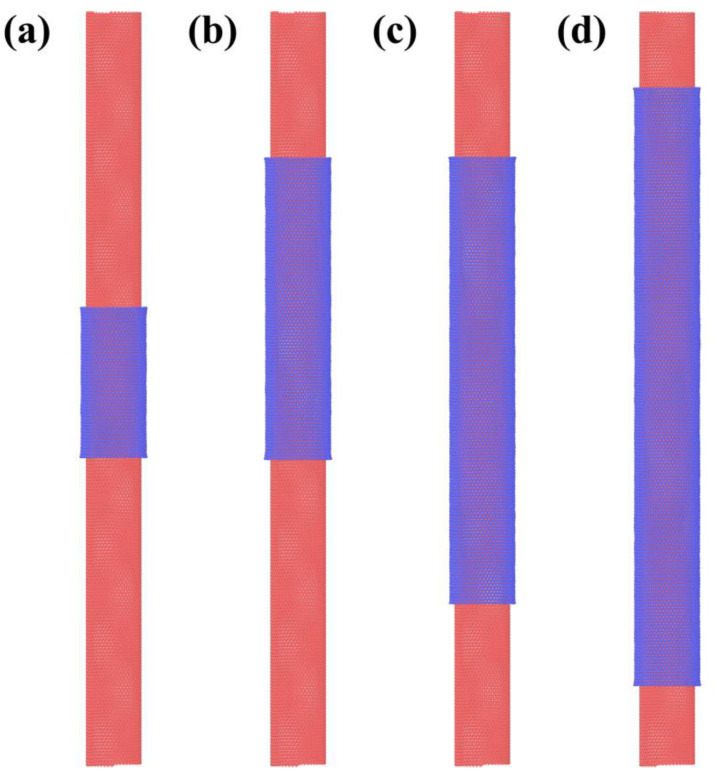
Wrapping (44,1) SWCNT with GNR of L = 13.38 nm, while W equals to 10.0, 20.0, 30.0 and 40.0 nm respectively from (**a**–**d**). The resultant twist angle is 1.1°.

**Figure 5 materials-15-08220-f005:**
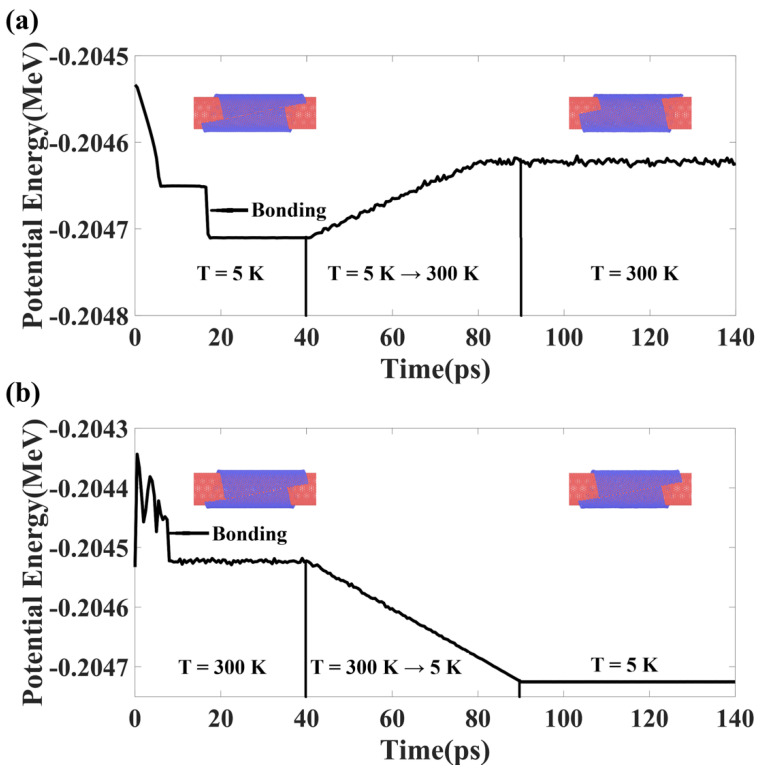
Influence of temperature environment on the dynamics of the self-assembled structure. (**a**,**b**) Two difference thermal schemes are adopted to the assemble process, both yield the same eventual configuration, which is found to be stable without plastic phase transformation. The (44,6) SWCNT and a GNR of 13.4 × 10.0 nm^2^ are used in this study.

**Figure 6 materials-15-08220-f006:**
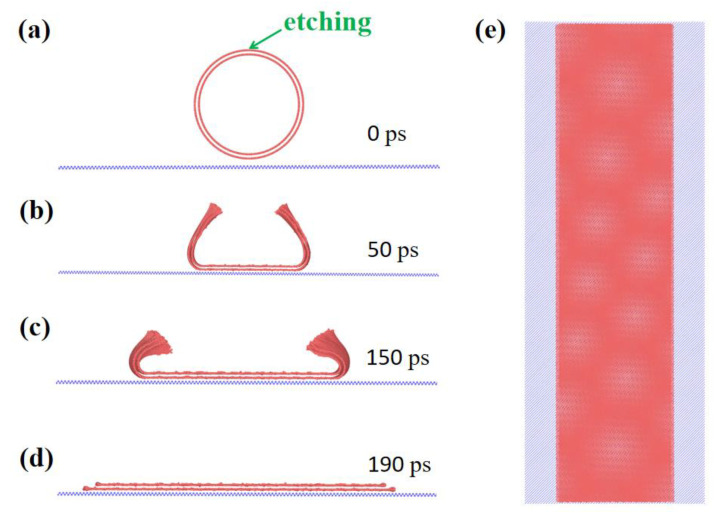
Unzipping a twisted (91,18)/(97,22) SWCNT via etching on substrate at 1 K, the twist angle is 1.14° and become 1.16° after collapsing onto the substrate. The snapshots are given in (**a**–**d**), the top view of the yielded moire pattern is shown in (**e**). The size of the unzipped bilayer graphene is 24.88 nm × 100 nm and 26.95 nm × 100 nm.

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
