# Peer review of "One Dimensional Twisted Van der Waals Structures Constructed by Self-Assembling Graphene Nanoribbons on Carbon Nanotubes"

_materials, 2022, doi:10.3390/ma15228220_

Round 1
Reviewer 1 Report
Review of 1D twisted…..
This is an extraordinary paper. Comments are below.
1. On line 29 can the magnetic properties of CNT be briefly explained?
2. Grammar could be improved in various places in the paper.
3. The short 22 nm nanotube length seems will be difficult to put into practical applications. Can longer GNR be used on mm long or larger diameter CNT? How will the CNT with graphene nanoribbons be connected to an electrical circuit? How do you envision the CNT wrapped with GNR will be used to make superconductivity devices? Can macroscale electronic devices be built using CNT wrapped with GNR?
4. How can the CNT with GNR be manufactured at large scale?
5. Some references have the year bold and some do not.
6. Besides wrapping CNT with GNR, are there other geometries that might be useful, such as putting aligned CNT between graphene sheets, or putting buckyballs between graphene sheets?
7. Millie Dresselhaus talked about the property of any DWNT by itself varies upon rotation and translation of the inner tube relative to the outer tube. Thus, can you achieve the properties of CNT wrapped graphene by appropriately translating and rotating the inner tube in a DWNT?
8. Reading the paper requires a background in molecular dynamics. Can the effects predicted by the simulations be explained in simpler terms for non expert readers?
9. Can you speculate what experiments could be tried to produce macroscale superconductive materials based on CNT wrapped with GNR?
Author Response
Reviewer #1
This is an extraordinary paper. Comments are below.
Comment#1: On line 29 can the magnetic properties of CNT be briefly explained?
Our Response:
First of all, we thank the reviewer for the positive evaluation on our work. As for the magnetic properties of CNTs, there are quite some different aspects, for example, Lu et al showed that when applying magnetic field to carbon nanotubes, it can drive metal-to-insulator transition (Lu, Phys. Rev. Lett., 74, 1123 (1995).). Ajiki showed that magnetic flux in CNTs can vary with different CNT structures (Ajiki et al. J. Phys. Soc. Japan. 62, 2470 (1993)). While the detail is not the central topic of our manuscript, we therefore did not include such discussions in the introduction part of our manuscript.
Comment#2: Grammar could be improved in various places in the paper.
Our Response:
We apologize for the grammatic errors we made in the original manuscript, we have now carefully read the manuscript and corrected them in the revision.
Comment#3: The short 22 nm nanotube length seems will be difficult to put into practical applications. Can longer GNR be used on mm long or larger diameter CNT? How will the CNT with graphene nanoribbons be connected to an electrical circuit? How do you envision the CNT wrapped with GNR will be used to make superconductivity devices? Can macroscale electronic devices be built using CNT wrapped with GNR?
Our Response:
The reviewer has raised important issues towards the applications of the present strategy which we achieved via simulations. Our understanding is that the 22nm nanotube is of course hard to be manipulated in real experiments, longer CNTs are more likely to be useful in device fabrications, we believe the underlying mechanism remain unchanged since it is shown that van der Waals interactions play key roles in the process. While connecting either CNT or GNRs to electric circuit is achievable in experimentalists, such as Cao et al’s approach. And early experiments by the Hongjie Dai group (Nature 424, 654-657 (2003); Phys. Rev. Lett. 100, 206803 (2008)) of Stanford University have also demonstrated both CNTs and GNRs can be well connected. As for macroscale electronic devices using CNT wrapped with GNR, we would expect this is achievable, however, it is relative expensive since producing such structures requires high technique and extreme synthesizing conditions, at current, not many labs can turn this in reality. We have added corresponding discussions in the revised manuscript.
Comment#4: How can the CNT with GNR be manufactured at large scale?
Our Response:
There are quite some progresses in recent years in producing both CNTs and GNRs with high yield. When putting together, experimentalist also have their innovative strategies. One of the examples is dispersing CNTs and GNRs separately in solution and then using coaxial syringe to eject them simultaneously and the wrapping process can take place spontaneously. Please refer to the literature (Tang et al. J. Solid. Stat. Chem. 224, 94 (2015)) for details and we have now added corresponding discussions in the revised manuscript.
Comment#5: Some references have the year bold and some do not.
Our Response:
We thank the reviewer for pointing out this issue, we have now adjusted the fonts of the references.
Comment#6: Besides wrapping CNT with GNR, are there other geometries that might be useful, such as putting aligned CNT between graphene sheets, or putting buckyballs between graphene sheets?
Our Response:
This is really an interesting suggestion, we actually have achieved one of the ideas that by putting buckyballs on the surface of CNTs and then wrap them with graphene sheets, the resultant structure was demonstrated to have great photovoltaic effect. Please refer to our previous publication for details (Tang et al. J. Solid. Stat. Chem. 224, 94 (2015)).
Comment#7: Millie Dresselhaus talked about the property of any DWNT by itself varies upon rotation and translation of the inner tube relative to the outer tube. Thus, can you achieve the properties of CNT wrapped graphene by appropriately translating and rotating the inner tube in a DWNT?
Our Response:
We thank the reviewer for bringing this message to our attention. While for DWCNTs, to pull the inner tube out from the outer sheath is extremely difficult since to locate the end of an inner tube requires very precise manipulation technique, while rotating is relatively easier to achieve because one can coat a pendulum to the outer tube and then rotate the pendulum. This technique has been demonstrated by Alex Zettl’s group from UC Berkeley. Please refer to the following literature for details (Nature, 424, 408–410 (2003)).
Comment#8: Reading the paper requires a background in molecular dynamics. Can the effects predicted by the simulations be explained in simpler terms for non expert readers?
Our Response:
We have modified some descriptions in the manuscript in order to make the presentations clear.
Comment#9: Can you speculate what experiments could be tried to produce macroscale superconductive materials based on CNT wrapped with GNR?
Our Response:
The reviewer has raised a very insightful question regarding future development of this area. As we have discussed in our response to reviewer comment # 3, the approach at nanoscale is more likely to happen since synthesizing macroscale structures of both CNTs and GNR is challenging. Once this challenge is tackled, we believe the same approach as we have discussed above would be useful.
Reviewer 2 Report
In this account paper, Zhou and coworkers have proposed a strategy to fabricate 1D van der Waals heterostructure via self-assembly of GNR onto the surface of SWCNT based on simulations. This topic would be attractive to the community, and the discussion is well-organized. However, there are still some points in the text need to be revised and clarified. Here I would put forwards my comments about this manuscript point by point:
1. Some representative references are suggested to be added, such as Zhao, Bei, et al. "High-order superlattices by rolling up van der Waals heterostructures." Nature 591.7850 (2021): 385-390. Yu, Decai, and Feng Liu. "Synthesis of carbon nanotubes by rolling up patterned graphene nanoribbons using selective atomic adsorption." Nano letters 7.10 (2007): 3046-3050. Lim, Hong En, et al. "Growth of carbon nanotubes via twisted graphene nanoribbons." Nature communications 4.1 (2013): 1-7.
2. When the twist angle between GNR and SWCNT is different, is the van der Waals interaction between GNR and CNT the same?
3. As for Phase I (L), with the increase of wrapped GNR layers, the radius of the nanotube also increases, has this variate been taken into account in the simulation?
Author Response
In this account paper, Zhou and coworkers have proposed a strategy to fabricate 1D van der Waals heterostructure via self-assembly of GNR onto the surface of SWCNT based on simulations. This topic would be attractive to the community, and the discussion is well-organized. However, there are still some points in the text need to be revised and clarified. Here I would put forwards my comments about this manuscript point by point:
Comment#1: Some representative references are suggested to be added, such as Zhao, Bei, et al. "High-order superlattices by rolling up van der Waals heterostructures." Nature 591.7850 (2021): 385-390. Yu, Decai, and Feng Liu. "Synthesis of carbon nanotubes by rolling up patterned graphene nanoribbons using selective atomic adsorption." Nano letters 7.10 (2007): 3046-3050. Lim, Hong En, et al. "Growth of carbon nanotubes via twisted graphene nanoribbons." Nature communications 4.1 (2013): 1-7.
Our Response:
We thank the reviewer for the positive comments on the scientific merit of our work, we also appreciate the reviewer for providing valuable information for improving our manuscript. The references the reviewer recommended are indeed relevant to the topic of our research. We have now added them into the corresponding discussions.
Comment#2: When the twist angle between GNR and SWCNT is different, is the van der Waals interaction between GNR and CNT the same?
Our Response:
The van der Waals interaction is very sensitive to the relative positions between atoms in the GNR and the SWCNT, the exact value between each pair is absolutely different when the twist angle changes. This effect, however, is more important if one graphene layer is stacked on the other layer with flat geometry, this is because in the study we presented, van der Waals interaction plays a role of driving GNR to attach to the surface of the CNT, once wrapped, the edge interaction determines the twist angle.
Comment#3: As for Phase I (L), with the increase of wrapped GNR layers, the radius of the nanotube also increases, has this variate been taken into account in the simulation?
Our Response:
The reviewer is absolutely correct about that with the increase of wrapped GNR layers, the radius of the nanotube also increases. We indeed have considered simulations with different CNT radius, part of the results are shown in Fig.3. This is also consistent with the observation in Fig.2 that when keep CNT radius constant, the twist angle varies with GNR length. A recent work from our group has offered a phase diagram regarding this issue (Sun et al. Acta Mech. Solida Sin. 2022, accepted, https://doi.org/10.1007/s10338-022-00358-9). The focus of that work is not on the twisting of GNR on CNT, as has been examined here, we therefore did not concentrate on discussing the radius effect in this manuscript.

Reviewer 3 Report
In their manuscript, Zhou et al use molecular dynamics simulations to show that graphene nanoribbons can self-assemble on the surface of carbon nanotubes, forming new heterostructures, possibly with a 1D moiré pattern. Overall, the results of this study are well presented and will be interesting for the community. However, there are some questions that need to be addressed before the manuscript can be considered for publication.
1) There are a few papers that cover a very similar topic and yet are not cited in the current manuscript:
DOI 10.1088/0031-8949/89/04/044008
DOI 10.1016/j.commatsci.2017.03.047
DOI 10.1021/jp205210x
The authors need to discuss these studies and how exactly they are related to this work.
2) It was shown previously that the Lenard-Jones potential does not capture variation of the binding energy upon shifting two graphene layers with respect to each other, so it may produce incorrect results in this study. A better alternative is the Kolmogorov-Crespi potential [DOI 10.1103/PhysRevB.71.235415], which is also implemented in LAMMPS. The authors need to rerun some simulations with this potential to check if their core predictions are reproduced with a more accurate potential.
3) Figure 1e needs to be improved, because right now it is difficult to see the difference between some configurations. One way to do this would be to add cross sections of each heterostructure.
4) When a nanoribbon wraps itself around the nanotube, is it possible that the system starts rotating around its axis as a whole? The authors should check if this happens in their calculations. If this is the case, it would be instructive to calculate the energy associated with this kind of rotation.
5) In line 101 and in other occasions, ‘2R’ needs to be replaced with ‘2\pi R’.
6) Line 84 has a typo in ‘heterostructuers’.
Author Response
In their manuscript, Zhou et al use molecular dynamics simulations to show that graphene nanoribbons can self-assemble on the surface of carbon nanotubes, forming new heterostructures, possibly with a 1D moiré pattern. Overall, the results of this study are well presented and will be interesting for the community. However, there are some questions that need to be addressed before the manuscript can be considered for publication.
Comment#1:There are a few papers that cover a very similar topic and yet are not cited in the current manuscript:
DOI 10.1088/0031-8949/89/04/044008
DOI 10.1016/j.commatsci.2017.03.047
DOI 10.1021/jp205210x
The authors need to discuss these studies and how exactly they are related to this work.
Our Response:
We thank the reviewer for the recognition of the value of our work to the community, we also found the reviewer’s comments valuable and we have performed additional simulations to address the reviewer’s concerns, which will be discussed below. The references the reviewer provided is very helpful for us to have a better understanding of what else is going on in this community. We have cited them and made additional discussions in the revised manuscript.
Comment#2: It was shown previously that the Lenard-Jones potential does not capture variation of the binding energy upon shifting two graphene layers with respect to each other, so it may produce incorrect results in this study. A better alternative is the Kolmogorov-Crespi potential [DOI 10.1103/PhysRevB.71.235415], which is also implemented in LAMMPS. The authors need to rerun some simulations with this potential to check if their core predictions are reproduced with a more accurate potential.
Our Response:
The reviewer is absolutely correct regarding the differences between different potentials for describing van der Waals interactions. We actually have used another recently developed potential, the ILP potential as a benchmark to validate our results from LJ potential. Following the reviewer’s suggestion, we have now tested the KC potential as well. We summarized our results in Fig. R1, as shown below. The good news is that from all the 3 potentials, we obtained almost the same final configurations and twist angles. As have been discussed in our response to reviewer#2’s 2nd comment, the role of van der Waals interaction is to drive GNR to the surface of CNT to wrap around, while the eventual configuration depends more on the edge states and the sizes of the GNRs and CNTs. We therefore argue that the main physics and conclusions are not influence here for our specific topic. We have added necessary discussions in our revision.
Fig. R1. Final configurations of GNRs wrapping on SWCNTs by using KC, LJ and ILP potentials for describing van der Waals interactions. Most of the configurations are almost exactly the same and the twist angles are the same, indicating the robustness of the assemble process that is not sensitive to the choice of potential. The only one exception is for the L=13.2 nm case where the bonding configurations are slightly different from others, this can be attributed to the numerical errors during simulations. The overall physical picture is not influenced.
Comment#3: Figure 1e needs to be improved, because right now it is difficult to see the difference between some configurations. One way to do this would be to add cross sections of each heterostructure.
Our Response:
We thank the reviewer for the nice suggestion, we have now modified Fig.1 e for better readership.
Comment#4: When a nanoribbon wraps itself around the nanotube, is it possible that the system starts rotating around its axis as a whole? The authors should check if this happens in their calculations. If this is the case, it would be instructive to calculate the energy associated with this kind of rotation.
Our Response:
This is a very interesting idea, although we did not observe the rotation behavior in our work, it is however interesting to investigate the dynamics of this motion, as we have discussed in our response to reviewer#1’s comments No. 7, external stimulation can be applied to make the wrapped GNR rotate around. As for the energy variation, it will likely be periodic but with small vibrations, similar to that in the GHz DWCNT oscillators (Guo et al, Phys. Rev. Lett. 91, 125501 (2003)).
Comment#5: In line 101 and in other occasions, ‘2R’ needs to be replaced with ‘2\pi R’.
Our Response:
We thank the reviewer’s careful reading of our manuscript, we have now corrected the error.
Comment#6: Line 84 has a typo in ‘heterostructuers’.
Our Response:
Thanks again and we have corrected the typo.

Reviewer 4 Report
Zhou et al. studied the 1D twisted van der Waals structures constructed by self-assembling graphene nanoribbons on carbon nanotubes by molecular dynamics simulations. The manuscript is suggested to be accepted after the following issues are addressed.
1) The authors studied the SWCNTs of various chirality and GNRs of various sizes were chosen to study the self-assembly process. What’s about the MWCNTs
2) The authors studied the bilayer graphene, what’s about the momo-layer or multi-layer case
3) The authors should be used complete form and then abbreviations such as 2D, LAMMPS, OVITO, DWCNT etc.
4) Many spelling and formatting typos in this paper, and we hope the authors can check and revise them thoroughly
5) The authors mentioned that This is because when L is somewhat smaller than 2R, an H phase or W phase firstly form, afterwards the van der Waals interaction between the edges drives the GNR twisting around to approach each other as a path to reduce the potential energy. What happened in the case of L is a larger than 2R.
Author Response
Zhou et al. studied the 1D twisted van der Waals structures constructed by self-assembling graphene nanoribbons on carbon nanotubes by molecular dynamics simulations. The manuscript is suggested to be accepted after the following issues are addressed.
Comment#1: The authors studied the SWCNTs of various chirality and GNRs of various sizes were chosen to study the self-assembly process. What’s about the MWCNTs
Our Response:
We thank the reviewer for the support of our manuscript and the helpful comments for us to improve the presentation. We have carefully made revisions following the reviewer’s comments as highlighted in our revision. As for MWCNTs, it would be an interesting topic to explore, we did not consider this issue is because our original motive is to fabricate magic angle bilayer graphene which has been demonstrated by Cao et al to possess novel physical properties.
Comment#2: The authors studied the bilayer graphene, what’s about the momo-layer or multi-layer case
Our Response:
It is our opinion that mono-layer graphene’s physical property can be more conveniently tuned by topological defects, while multi-layer graphene offers more room for tuning inter-layer interactions and therefore their potential applications. We thank the reviewer bring out this topic and we surely will start investigating this issue ASAP and report the results once it is completed. As for now, we are focusing on bilayer graphene related topic owing to the magic angle physics, which is very urgent to tackle.
Comment#3: The authors should be used complete form and then abbreviations such as 2D, LAMMPS, OVITO, DWCNT etc.
Our Response:
We thank the reviewer for the careful readings of our manuscript, we have made corresponding revisions carefully.
Comment#4: Many spelling and formatting typos in this paper, and we hope the authors can check and revise them thoroughly
Our Response:
Thanks again and we have tried our best to correct these errors.
Comment#5: The authors mentioned that This is because when L is somewhat smaller than 2R, an H phase or W phase firstly form, afterwards the van der Waals interaction between the edges drives the GNR twisting around to approach each other as a path to reduce the potential energy. What happened in the case of L is a larger than 2R.
Our Response:
This is a very important question, we have now provided cross-sectional images in the revised manuscript, Fig.1e, where it can be seen when L is larger than 2pR, the longitudinal wrapping continues and form scroll type structures. This structure has been discussed in one of our previous papers (Tang et al, Carbon 61, 458 (2015)).

Round 2
Reviewer 3 Report
The authors have implemented the required changes and the manuscript can now be accepted for publication.